# Preclinical and Preliminary Evaluation of Perceived Image Quality of AI-Processed Low-Dose CBCT Analysis of a Single Tooth

**DOI:** 10.3390/bioengineering11060576

**Published:** 2024-06-07

**Authors:** Na-Hyun Kim, Byoung-Eun Yang, Sam-Hee Kang, Young-Hee Kim, Ji-Yeon Na, Jo-Eun Kim, Soo-Hwan Byun

**Affiliations:** 1Department of Conservative Dentistry, Hallym University Sacred Heart Hospital, Anyang 14066, Republic of Korea; 2Department of Oral and Maxillofacial Surgery, Hallym University Sacred Heart Hospital, Anyang 14066, Republic of Korea; 3Graduate School of Clinical Dentistry, Hallym University, Chuncheon 24252, Republic of Korea; 4Institute of Clinical Dentistry, Hallym University, Chuncheon 24252, Republic of Korea; 5Dental AI-Robotics Center, Hallym University Sacred Heart Hospital, Anyang 14066, Republic of Korea; 6Department of Oral and Maxillofacial Radiology, Hallym University Sacred Heart Hospital, Anyang 14066, Republic of Korea; 7Department of Oral and Maxillofacial Radiology, Seoul Nation University Dental Hospital, Seoul 03080, Republic of Korea

**Keywords:** CBCT, cone-beam computed tomography, AI, artificial intelligence, image quality enhancement, protection, radiation

## Abstract

This study assessed AI-processed low-dose cone-beam computed tomography (CBCT) images for single-tooth diagnosis. Human-equivalent phantoms were used to evaluate CBCT image quality with a focus on the right mandibular first molar. Two CBCT machines were used for evaluation. The first CBCT machine was used for the experimental group, in which images were acquired using four protocols and enhanced with AI processing to improve quality. The other machine was used for the control group, where images were taken in one protocol without AI processing. The dose-area product (DAP) was measured for each protocol. Subjective clinical image quality was assessed twice by five dentists, with a 2-month interval in between, using 11 parameters and a six-point rating scale. Agreement and statistical significance were assessed with Fleiss’ kappa coefficient and intra-class correlation coefficient. The AI-processed protocols exhibited lower DAP/field of view values than non-processed protocols, while demonstrating subjective clinical evaluation results comparable to those of non-processed protocols. The Fleiss’ kappa coefficient value revealed statistical significance and substantial agreement. The intra-class correlation coefficient showed statistical significance and almost perfect agreement. These findings highlight the importance of minimizing radiation exposure while maintaining diagnostic quality as the usage of CBCT increases in single-tooth diagnosis.

## 1. Introduction

Accurate diagnosis leading to an appropriate treatment plan is crucial for successful dental treatment. Therefore, radiology is essential in dentistry for diagnostic purposes, planning and execution of treatment, and evaluation of therapeutic success [1]. Since 1895, radiographic imaging has become an increasingly important adjunct for diagnosing diseases and planning appropriate treatments [2]. However, conventional radiographs, such as periapical and panoramic views, provide only two-dimensional (2D) images of three-dimensional (3D) objects, which introduces limitations such as 3D compression of 2D images, necessitating multiple radiographs for a complete view [3,4]. Geometric alterations can distort images, thereby affecting diagnostic accuracy. Distortion levels vary, reaching >14% with orthopantomography [5]. Additionally, certain anatomical structures can create obstacles, obscuring the area of interest and complicating radiological interpretation [6].

Indiscriminate use of cone-beam computed tomography (CBCT) to obtain anatomical information for tooth diagnosis should be avoided. Initial clinical examination and conventional radiography, such as periapical images, should be conducted. CBCT should be performed only when the clinical benefits outweigh the risks associated with radiation exposure [7]. In relation to single-tooth diagnosis, suitable field of view (FOV) modes in CBCT may be considered in several situations. First, CBCT can detect pathological changes in periapical areas earlier than conventional 2D radiography [8,9]. Second, in cases where anatomical complexity presents challenges, such as dens invaginatus and severely curved root canals (e.g., radix entomolaris), pretreatment imaging is beneficial [10]. Third, CBCT can be considered in cases of severe avulsion or horizontal root fractures in teeth and surrounding tissues, where conventional evaluation methods such as periapical or occlusal radiographs are insufficient [11]. Fourth, CBCT imaging can be useful in cases of accidental perforation or internal resorption leading to root perforation [12]. Finally, when planning periapical surgery, 3D imaging is necessary to accurately understand the relationship between the periapex and vital anatomical structures [13]. CBCT can aid in assessing proximity to the inferior alveolar nerve, maxillary sinus, and other crucial structures during surgical planning [14].

However, CBCT also has disadvantages. It is more expensive and exposes the patient to higher levels of radiation than conventional radiography, and scatter and beam hardening that occur owing to high-density structures in the area of interest can reduce image quality [15].

Adjusting radiation dose exposure from radiological examinations “as low as reasonably achievable” (ALARA) has been extensively studied [16]. Once the ideal FOV is established for a patient, lower radiation doses can be obtained for CBCT scans. Reducing the tube current (mA), scan time (s), resolution (i.e., increasing voxel size), and number of projections and/or using a partial rotation mode (e.g., 180° instead of 360° rotation) are general ways to achieve so-called low-dose procedures. However, reducing the radiation dose in CBCT is indivisibly linked to image quality deterioration owing to increasing image noise. The limitations of conventional reconstruction methods for image quality enhancement in low-dose computed tomography (CT) have been previously explored. However, more recently, the advent of artificial intelligence (AI)-based post-processing denoising solutions has shown promising results in further enhancing image quality [17,18,19].

The rapid growth of deep learning, particularly convolutional neural networks (CNNs), has revolutionized CBCT procedures by reducing radiation risks. CNNs automatically update features using input data and enhance image analysis accuracy for early detection of abnormalities [20]. Deep neural networks (DNNs) are used to process complex CBCT data and aid in precise anatomical segmentation, which is crucial for treatment planning. Generative adversarial networks (GANs) contribute to data augmentation by generating realistic synthetic CBCT images. The integration of a trainable Sobel convolution into an edge enhancement-based densely connected CNN (EDCNN) is a notable advancement, providing a potent framework for image processing. The synergy between the trainable Sobel convolution and EDCNN enables end-to-end image denoising, with a variable Sobel factor enhancing edge information extraction during training [21,22].

Therefore, the aim of the present study was to evaluate the clinical efficacy, radiation exposure, and subjective image quality of AI-processed low-dose CBCT images for the diagnosis of a single tooth. In this study, general CBCT was used as the control group. Experimental group images, specifically using low-dose CBCT, were captured using four distinct combinations of filter/non-filter and standard/fast modes. Each image was processed by AI. The null hypothesis was that the type of CBCT and whether AI processing was applied would not affect the subjective image evaluation results.

## 2. Materials and Methods

### 2.1. CBCT Scanning Protocol

Human-equivalent phantoms provided by Erler Zimmer Co. (Lauf, Germany) were used to evaluate image quality. CBCT images of the right mandibular first molars of the phantoms were obtained. All scans were performed at Hallym University Sacred Heart Hospital. The phantom skull was set up on a tripod and, owing to its limited FOV, it was positioned to focus on the right mandibular first molar.

Two CBCT scanning machines were used in this study. An ASAHI ALPHARD (Asahi Roentgen, Kyoto, Japan) was used for the control group, while a Bright mCT (Dentium, Suwon, Republic of Korea) was used for the experimental group (Figure 1).

CBCT images of the experimental group were captured using four different combinations of filter/non-filter and standard/fast modes. The only difference between the filter and non-filter modes was the application of the filter, while the voxel size (µm) and exposure time (s) differed between the standard and fast modes. The voxel size for the standard mode was 249 µm, while that for the fast mode was 390 µm. The exposure time for the standard mode was 20 s, whereas that for the fast mode was 10 s. In the filter mode, a filter was placed in the CBCT radiation-generating unit to reduce the amount of radiation exposure to the patient. The standard mode without the filter was labeled as protocol 1, the fast mode without the filter was labeled as protocol 2, the standard mode with the filter was labeled as protocol 3, and the fast mode with the filter was labeled as protocol 4. Protocols 1–4 shared the same FOV of 10 × 8 cm^2^, tube voltage of 95 kVp, and tube current of 10 mA. All images underwent AI processing. In the control group, a single protocol was employed for imaging, referred to as protocol 5 in this article. Protocol 5 had an FOV of 5 × 5 cm^2^, voxel size of 100 µm, exposure time of 17 s, tube voltage of 80 kVp, and tube current of 8 mA. The details of the CBCT scanning protocol, such as FOV (cm × cm), voxel size (µm), exposure time (s), tube voltage (kVp), and tube current (mA), are described in Table 1. Figure 2 shows the CBCT images captured using protocols 1–5.

### 2.2. Dose-Area Product Measurement

Dose-area product (DAP) was measured by attaching an ion chamber of a DAP meter (KERMAX^®^ plus DDP TinO, IBA DOSIMETRY, Schwarzenbruck, Germany) to the center of the radiographic tube for each of setting of protocols 1–5. The process involved three repetitions of the measurements and the calculation of the average value for each protocol.

### 2.3. AI Processing

An EDCNN was used for the denoising of low-dose CT images [23]. Figure 3 depicts the architecture of the network. First, the edge enhancement module consisting of the trainable Sobel convolutions (eight groups of the four types) extracted edge information and enriched the input information. The Sobel convolution was designed with a learnable Sobel parameter in the form of the traditional Sobel operator. Accordingly, 32 different edge features were extracted using this module. At the last stage of the module, low-dose CT images were stacked with the extracted edge features in the channel dimension. Next, the dense connection module was composed of 1 × 1 and 3 × 3 convolutions and Leaky ReLU layers. To maximize the use of the extracted edge features and original input, edge features extracted from the edge enhancement module were connected by skip connection to each B module. The output channel of the module remained consistent at 32. Through point-wise convolution with a 1 × 1 kernel, the channel size was converted to 1, aligning it with the input channel. The convolutional network with a padded 3 × 3 kernel was used to adjust the output spatial size to the input size. Finally, denoised images were obtained by adding the original low-dose CT images via skip connection.

### 2.4. Subjective Clinical Image Quality Evaluation

In the experimental group, CBCT images were obtained following protocols 1–4, all of which underwent AI processing, resulting in eight CBCT images centered around the right mandibular first molar. The control group, referred to as protocol 5, had one CBCT image centered around the right mandibular first molar. Therefore, a total of nine images obtained using different protocols were randomly presented for subjective clinical image quality evaluation. Three conservative dentistry specialists and two oral and maxillofacial (OMF) radiology specialists participated in the evaluation. The evaluation criteria comprised 11 parameters, as follows: (1) number of roots, (2) number of root canals, (3) enamel–dentin differentiation, (4) lamina dura, (5) periodontal ligament (PDL) space, (6) trabecular pattern, (7) cortex of the alveolar crest, (8) cortex of the mandibular canal, (9) furcation, (10) cortex of the mandible, and (11) overall image quality for periapical lesion diagnosis. Participants rated their responses on a six-point scale: (1) strongly disagree, (2) disagree, (3) slightly disagree, (4) slightly agree, (5) agree, and (6) strongly agree. Images were presented to the evaluators using the Rainbow 3D viewer 1.0.0 (Dentium, Suwon, Republic of Korea), which allows axial, coronal, and sagittal views of CBCT images for evaluation (Figure 4). This is the same program used to capture actual CBCT images in clinical settings and for diagnostic evaluation. To ensure fairness, an observation time of 3 min was allotted for each image. To investigate intra-rater reliability, subjective clinical image quality evaluations were performed twice by the same evaluators, with a 2-month interval in between, to allow for a washout period.

### 2.5. Statistical Analyses

SPSS (version 20, IBM Corp, Armonk, NY, USA) was used for statistical analyses. Inter-reader reliability of parameter average was assessed using Fleiss’ kappa statistics. The Fleiss’ kappa coefficient was applied to assess agreement and statistical significance (<0.00, poor agreement; 0.00–0.20, slight agreement; 0.21–0.40, fair agreement; 0.41–0.60, moderate agreement; 0.61–0.80, substantial agreement; and 0.81–1.00, almost perfect agreement). The Fleiss’ kappa coefficient > 0 indicates progressively better-than-chance agreement among raters, with a maximum value of +1 signifying perfect agreement. *p* < 0.05 was considered to indicate statistical significance [24,25]. Intra-rater reliability was assessed using intra-class correlation coefficient (ICC). A six-tiered classification system was utilized to evaluate the level of agreement: poor agreement (ICC, <0.00), slight agreement (ICC, 0.00–0.20), fair agreement (ICC, 0.21–0.40), moderate agreement (ICC, 0.41–0.60), substantial agreement (ICC, 0.61–0.80), and almost perfect agreement (ICC, 0.81–1.00). All *p* values < 0.05 were considered statistically significant [26].

## 3. Results

### 3.1. DAP Measurement

Among the five protocols, the lowest DAP and DAP/FOV values were observed with protocol 4, which corresponded to the fast mode with the filter applied. The values for this protocol were 27.573 µGy·m^2^ and 0.335 µGy·m^2^/cm^2^, respectively. The second-lowest DAP and DAP/FOV values were associated with protocol 3, the standard mode with the filter applied, with values of 55.49 µGy·m^2^ and 0.694 µGy·m^2^/cm^2^, respectively. In contrast, the highest DAP and DAP/FOV values were observed with protocol 1, corresponding to the standard mode without the filter. The respective values for this protocol were 261.503 Gy·m^2^ and 3.269 Gy·m^2^/cm^2^. The average DAP and DAP/FOV values for all protocols used in this study are described in Table 2.

### 3.2. Subjective Image Quality Evaluation

The average values of the participants’ parameter evaluations for the given images are listed in Table 3. The results of the subjective clinical evaluations indicated that AI-processed protocol 1 received the highest score, followed by AI-processed protocols 2 and 5. The lowest score was attributed to protocol 4 images without AI processing (score order: 1 AI, 2 AI, 5, 1, 3 AI, 2, 4 AI, 3, and 4). Incorporating AI processing into the experimental group images resulted in an overall enhancement of the subjective clinical evaluation scores. AI-processed protocol 2 images showed higher scores than non-processed protocol 1 images. AI-processed protocol 3 images showed higher scores than non-processed protocol 2 images. AI-processed protocol 4 images showed higher scores than non-processed protocol 3 images. Even though AI-processed protocol 2 images had lower DAPs per FOV than protocol 5 images, AI-processed protocol 2 images showed higher scores than protocol 5 images (Table 3). The Fleiss’ kappa coefficient of parameter averages was 0.621, which was statistically significant (*p =* 0.030). The Fleiss’ kappa coefficient showed substantial agreement. Additionally, the ICC value of 0.995 indicated an almost perfect agreement in intra-rater reliability. The ICC value ranged from 0.976 to 0.998 within the 95% confidence interval. The *p*-value was <0.001, demonstrating statistically significant results.

## 4. Discussion

Dentists should possess sufficient information about CBCT before performing scans, and knowledge of radiation protection is essential. The amount of radiation exposure to the human body is typically evaluated in terms of the effective absorbed dose. According to several studies, the effective absorbed radiation dose from CBCT is significantly lower than that from conventional CT [27]. However, it has been reported to be considerably higher than those of panoramic and periapical images [15]. Therefore, dentists must engage in thorough discussions and obtain consent from patients before CBCT imaging, adhering to the ALARA principle to minimize radiation dosage. This radiation dose reduction is vital to promote patient safety and minimize the difficulty of predicting long-term harm, especially for patients with systemic diseases and young patients in the growth stage [28]. Various methods can be employed to reduce radiation dose, including reducing the tube current (mA), scan time (s), resolution (i.e., increasing the voxel size), and number of projections and/or using a partial rotation mode (e.g., 180° instead of 360° rotation). These approaches result in what is commonly referred to as low-dose CBCT. In this study, the experimental group achieved a reduction in DAP/FOV (µGy·m^2^/cm^2^) by applying filters, increasing voxel size, and decreasing exposure time. However, radiation dose reduction is indivisibly linked to image quality deterioration owing to increased image noise [17]. In this study, the experimental group achieved a reduction in the radiation dose using the aforementioned methods. However, because an inevitable deterioration in image quality was anticipated, AI processing was implemented.

AI, which has been applied in a broad range of industries in recent years, is an active area of interest for many researchers. Dentistry is no exception to this trend, and the applications of AI are particularly promising in the field of OMF radiology. Recent research on AI in OMF radiology has primarily employed CNNs, which can perform image classification, detection, segmentation, registration, generation, and refinement. AI systems have been developed for radiographic diagnosis, image analysis, forensic dentistry, and image quality improvement [29]. Several deep learning algorithms have been developed for low-dose CBCT, and various networks have been derived from U-shaped architectures, deep CNNs, GANs, variational autoencoders, and deep residual networks. Moreover, synthetic CT images can be generated from magnetic resonance imaging using a CNN without acquiring actual CT images. A similar approach can be used to improve the quality of low-dose CT images to the level of high-resolution CT images. Several studies have found that the image quality of deep learning-based synthesized CT has improved overall, with considerably lower mean absolute error differences, compared with the reference CT in both the testing and validation datasets. According to Zhang et al., a deep learning algorithm is promising for efficiently improving CBCT image quality; thus, it has the potential to support online CBCT-based adaptive radiotherapy [15,30].

The deep learning algorithm EDCNN was used in the present study. With the aid of specialized learning technologies, the EDCNN was developed with a deep multilayer perceptron, capable of creating a safe and improved classification model with nonlinear and linear functions, regularization, falling, and binary sigmoid classifications. Consequently, deep learning prediction models and classification can provide highly accurate and reliable evaluations while reducing the incidence of misdiagnoses. According to Yan et al., compared with existing low-dose CT image denoising algorithms, the EDCNN model proposed in this study better preserves details and suppresses noise [31].

This study compared subjective image quality by obtaining CBCT images of a phantom skull. The various parameters for the evaluation of subjective image quality were selected for the following reasons. During the diagnosis of a single tooth, important information is obtained by assessing the number of roots and root canals. CBCT facilitates the easy detection of additional anatomical features such as the second mesiobuccal canal in maxillary molars, additional roots (e.g., radix entomolaris), or the C-shaped canal system in mandibular molars. Assessing endo-origin lesions is also crucial, where changes occur in various anatomical structures, such as the enamel–dentin junction, lamina dura, PDL space, trabecular pattern, cortex of the alveolar crest, cortex of the mandibular canal, furcation, and cortex of the mandible, if inflammation is present. The results of the subjective clinical evaluations showed that the AI-processed images in protocol 1 received the highest score, and the AI-processed images of protocol 2 had the second-highest scores. These subjective image evaluation scores were higher than those of the control group. Therefore, the null hypothesis that CBCT type and AI processing do not affect subjective image evaluation results was rejected. However, in protocols 3 and 4, even with AI processing after applying filters, the subjective image evaluation results were inferior to those of the control group. In addition, this study used subjective image quality evaluation to evaluate each protocol. Since there is no way to objectively and accurately judge image quality, we believe that it is meaningful for clinicians who evaluate radiographic images in clinical practice to evaluate each protocol by themselves. This study used Fleiss’s kappa statistics to check inter-reader reliability, which revealed that the clinicians who participated in the evaluation showed statistically significant, substantial agreement. This study also used ICC statistics to assess intra-rater reliability, which showed statistical significance and almost perfect agreement. These results show that the subjective image quality evaluation in this study did not differ significantly in the inter- and intra-rater evaluations.

The DAP/FOV ratio for AI-processed images of protocol 1, which exhibited the highest subjective image evaluation results, was 3.269 µGy·m^2^/cm^2^, higher than that of the control group, images of protocol 5 (2.823 µGy·m^2^/cm^2^). Conversely, the DAP/FOV value for AI-processed images of protocol 2, which showed the second-best subjective image evaluation results, was 1.634 µGy·m^2^/cm^2^, lower than that of the control group, images of protocol 5. Therefore, in this study, AI-processed images of protocol 2 demonstrated lower DAP/FOV values while exhibiting subjective clinical evaluation results comparable to those of conventional CBCT, indicating its clinical relevance. In addition, the subjective image evaluation in the experimental group showed that AI-processed protocol 2 images showed a higher score than non-processed protocol 1 images. AI-processed protocol 3 images showed a higher score than non-processed protocol 2 images. AI-processed protocol 4 images showed a higher score than non-processed protocol 3 images. This confirms that AI processing can improve radiological image quality by more than one level.

Recent surveys have found that half of the endodontists in the United States utilize CBCT devices in their practices [32]. Among various indications, >50% of endodontists have discovered missed canals using CBCT images (“occasionally”, “frequently”, or “always”). Similar surveys in the United Kingdom have also indicated that the evaluation of complex root canal systems is one of the most significant indications for CBCT use [33]. As CBCT use has recently increased in tooth diagnosis, the demand for high-resolution CBCT images to assess findings such as periapical lesions and root canal abnormalities with higher accuracy has grown [34]. However, using high-resolution CBCT images to obtain superior image quality can result in higher radiation doses, particularly in sensitive areas such as the eyes and thyroid gland, emphasizing the importance of efforts to reduce radiation exposure [35]. The findings of this study are significant, as AI-processed low-dose CBCT in the experimental group demonstrated subjective image quality evaluation results that were clinically comparable to those of conventional CBCT in the diagnosis of a single tooth. However, further developing AI-processing technology to improve image quality is important.

This study had several limitations. First, the number of subjective clinical evaluators was limited. More reliable results could be obtained with a greater number of participating clinical evaluators. However, the scoring propensities of the evaluators who participated were similar, and the results are not expected to change significantly if the number of evaluators increased. Moreover, inter- and intra-reliability assessments were performed in this study to ensure the reliability of subjective ratings. While this may still be insufficient, we acknowledge it as an unavoidable limitation of the method used to assess actual clinical effectiveness. Second, because a phantom skull was used, actual pathologies, such as apical lesions, or anatomical structures, such as canal calcifications, were not present. Obtaining images from a diverse range of cases involving real patients would lead to more clinically meaningful results. However, ensuring the standardization of examination can be difficult if the skull contains pathologic problems or abnormal anatomical structures. A phantom skull with normal findings would be more appropriate for evaluating the quality of radiographic images. Hence, a phantom skull was used in this study. Third, no motion was detected during CBCT scans of the phantom heads. However, even slight movements by patients can cause geometric errors in the reconstruction process, resulting in motion blurring and reduced spatial resolution. This may lead to more noticeable artifacts, such as double or multiple contours, and certain motion patterns may become apparent. Fourth, a discrepancy was observed in the FOV sizes between the experimental and control groups, making a completely accurate comparison impossible. Nonetheless, this study remains significant because it followed the ALARA principle to acquire clinically meaningful images necessary for the diagnosis of a single tooth with the minimum possible dose by utilizing AI processing.

## 5. Conclusions

This research concluded that incorporating AI into low-dose CBCT image processing is valuable in clinical practice, significantly enhancing subjective image quality. This study’s findings validate the use of low-dose CBCT for the diagnosis of a single tooth, relying on subjective image assessment with key parameters. By utilizing this method, precise diagnosis can be achieved with reduced radiation exposure, easing the burden of extensive examinations for patients, especially for those whom radiation exposure is a concern, such as adolescents and individuals with systemic conditions. These results can serve as a pivotal basis for future studies and contribute to the advancement of safe and efficient image enhancement through AI technologies, both in clinical practice and from an ethical perspective.

## Figures and Tables

**Figure 1 bioengineering-11-00576-f001:**
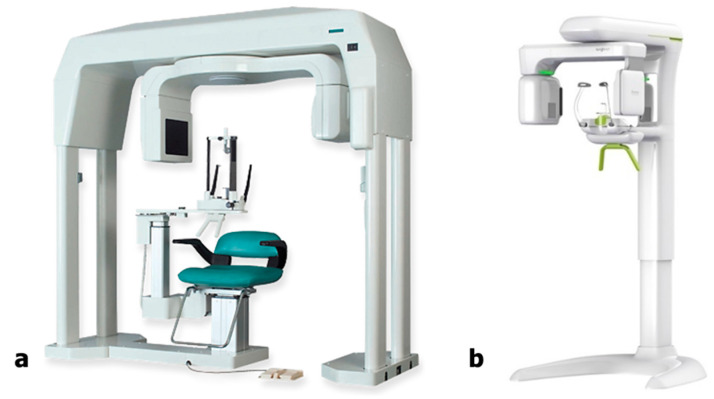
(**a**) ASAHI ALPHARD (Asahi Roentgen, Kyoto, Japan), used for the control group and (**b**) Bright mCT (Dentium, Suwon, Republic of Korea), used for the experimental group.

**Figure 2 bioengineering-11-00576-f002:**
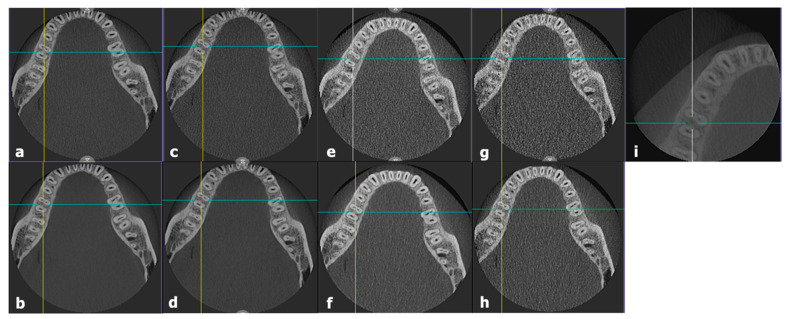
Cone-beam computed tomography images obtained using (**a**) protocol 1; (**b**) AI-processed protocol 1; (**c**) protocol 2; (**d**) AI-processed protocol 2; (**e**) protocol 3; (**f**) AI-processed protocol 3; (**g**) protocol 4; (**h**) AI-processed protocol 4; and (**i**) protocol 5.

**Figure 3 bioengineering-11-00576-f003:**
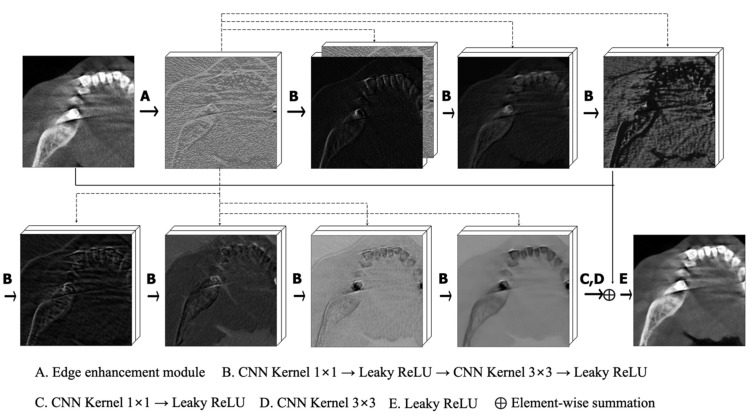
The overall architecture of the AI model employing the edge enhancement-based densely connected convolutional neural network (EDCNN).

**Figure 4 bioengineering-11-00576-f004:**
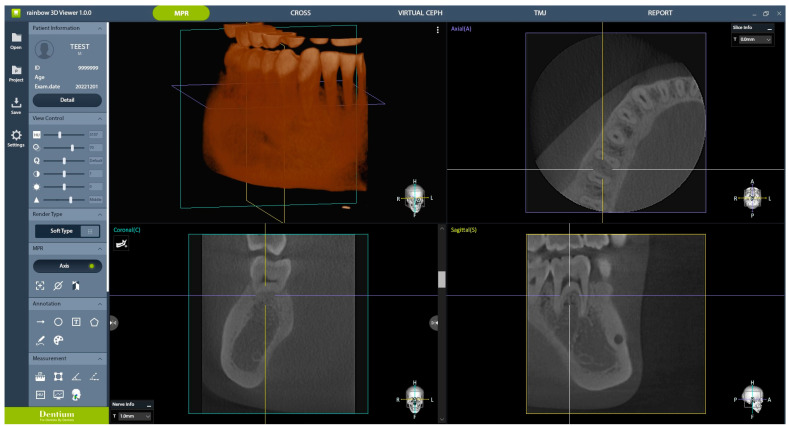
The images presented to the evaluators for subjective clinical image quality evaluation.

**Table 1 bioengineering-11-00576-t001:** The details of the cone-beam computed tomography scanning protocols.

Group	Mode	Protocol No.	FOV (cm^2^)	Voxel Size (µm)	Exposure Time (s)	kVp	mA
Experimental group	Non-filter	1	10 × 8	249	20	95	10
		2 *	10 × 8	390	10	95	10
	Filter	3	10 × 8	249	20	95	10
		4 *	10 × 8	390	10	95	10
Control group	Standard	5	5 × 5	100	17	80	8

* indicates fast mode; FOV, field of view.

**Table 2 bioengineering-11-00576-t002:** The average DAP and DAP/FOV values with protocols 1–5.

Model	Mode	Protocol No.	DAP (µGy·m^2^)	DAP per FOV (µGy·m^2^/cm^2^)
Experimental group (10 × 8)	Non-filter	1	261.503	3.269
		2 *	130.693	1.634
	Filter	3	55.49	0.694
		4 *	27.573	0.335
Control group (5 × 5)		5	70.575	2.823

* indicates fast mode; DAP, dose-area product; FOV, field of view.

**Table 3 bioengineering-11-00576-t003:** Results of the subjective image quality evaluation.

Protocol No.	1	1	2 *	2 *	3	3	4 *	4 *	5
Parameter	(AI)		(AI)		(AI)		(AI)		
Number of roots	5.625	5.75	5.5	5.375	5.375	4.75	4.25	3.5	5.5
Number of root canals	5.875	5.75	5.375	5.25	5.625	4.75	3.75	3.125	5.5
Enamel–dentin differentiation	5.125	5.625	5.5	4.375	4.375	4	3.875	2.75	4.375
Lamina dura	4.5	3.5	4.5	3.625	3.875	2.75	3.375	2	4
PDL space	5	4.625	4.5	4.5	4.125	3.625	3.375	2.375	5.25
Trabecular pattern	5.25	4	4.5	3.75	4.375	2.125	3.25	1.75	5.375
Cortex of alveolar crest	5.625	5	5.25	4.5	4.625	3.5	3.875	2.875	5.125
Cortex of mandibular canal	5.75	5.5	5.625	5.125	5.25	4	4.125	3.75	5.5
Furcation	5.5	5.5	5.25	4.5	5	3.75	3.75	3	5
Cortex of mandible	5.5	5.5	5.75	5.5	5.5	5	4.75	4.25	5
Overall image quality for PA lesion diagnosis	4.75	4.375	4.75	4	4	2.5	2.75	2.25	5.25
**Average**	5.32	5.01	5.14	4.59	4.74	3.71	3.74	2.86	5.08

* indicates fast mode; PDL, periodontal ligament; PA, periapical.

## Data Availability

Datasets supporting the conclusions of this study are included in the present article. The datasets used and/or analyzed in the current study are available from the corresponding author upon reasonable request.

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
