# Peer review of "Preclinical and Preliminary Evaluation of Perceived Image Quality of AI-Processed Low-Dose CBCT Analysis of a Single Tooth"

_bioengineering, 2024, doi:10.3390/bioengineering11060576_

Round 1

Reviewer 1 Report

Comments and Suggestions for Authors

Dear Authors,

Thank you very much for submitting your manuscript entitled „ Clinical effectiveness of AI-processed low-dose CBCT for endodontic treatment”.

The study aims at dose reduction and improvement of image quality and therefore serves for better diagnosis.

However, I have some major concerns.

The study is somehow a black box. You focussed on endodontic treatment and evaluated the following aspects:

The evaluation criteria comprised 11 parameters, as follows: (1) number of roots, (2) number of root canals, (3) enamel-dentin differentiation, (4) lamina dura, (5) periodontal ligament (PDL) space, (6) Trabecular pattern, (7) cortex of the alveolar crest, (8) cortex of the mandibular canal, (9) furcation, (10) cortex of the mandible, and (11) overall image quality for periapical lesion diagnosis. Participants rated their responses on a six-point scale: (1) strongly disagree, (2) disagree, (3) slightly disagree, (4) slightly agree, (5) agree, and (6) strongly agree.

To my point of view, that are not primary endodontic pattern and I miss a concrete endodontic diagnosis that should be present.

The use of phantom heads may be adequate, but what is about simulating moving artifacts and all the other artifacts that are patient associated?

I question clinical relevance, as the mandible did not offer any typical endodontic diagnosis.

Why was the evaluation limited to only one tooth? Why did you use different sizes of the FOV? You should better address the clinical necessity of larger FOVs – if present. I guess, the more information with the same or even lower doses, the better.

To my point of view, the study simple highlighted the benefit of some AI-processing procedures but did not aim at an endodontic topic. For me, it is a native mandible and the AI processing led to dose reduction – please, do not use endodontic diagnoses masking the primary aim.

The title should be changed and you should focus on the true aim – AI processing and dose reduction. The study does not answer the question of “Clinical effectiveness“, as mentioned in the title.

To prove clinical effectiveness, the AI processing should be performed in real clinical cases – this would be more relevant.

Recent publications addressed the topic even better:

Lee, J. H., Lee, N. K., Zou, B., Park, J. H., & Choi, T. H. (2024). Reliability of Artificial Intelligence-based Cone Beam Computed Tomography Integration with Digital Dental Images. Journal of visualized experiments : JoVE, (204), 10.3791/66014. https://doi.org/10.3791/66014

Kim, K., Lim, C. Y., Shin, J., Chung, M. J., & Jung, Y. G. (2023). Enhanced artificial intelligence-based diagnosis using CBCT with internal denoising: Clinical validation for discrimination of fungal ball, sinusitis, and normal cases in the maxillary sinus. Computer methods and programs in biomedicine, 240, 107708. https://doi.org/10.1016/j.cmpb.2023.107708

Kim, K. S., Kim, B. K., Chung, M. J., Cho, H. B., Cho, B. H., & Jung, Y. G. (2022). Detection of maxillary sinus fungal ball via 3-D CNN-based artificial intelligence: Fully automated system and clinical validation. PloS one, 17(2), e0263125. https://doi.org/10.1371/journal.pone.0263125

Thus, redundancies do not help clinicians.

Comments on the Quality of English Language

only minor editing is necessary

Author Response

Point-to-Point Response to the Comments of Reviewer (bioengineering-2974874)

We are grateful to the reviewers for their critical comments and valuable suggestions that have helped us improve our manuscript considerably. As indicated in the following responses, we have incorporated all these suggestions into the revised version of our paper. The edited sentences are written in red in the revised manuscript. The page and line numbers are indicated in accordance with the manuscript in the Microsoft Word file.

<Reviewer 1>

Comment #1:

The study is somehow a black box. You focussed on endodontic treatment and evaluated the following aspects: 
The evaluation criteria comprised 11 parameters, as follows: (1) number of roots, (2) number of root canals, (3) enamel-dentin differentiation, (4) lamina dura, (5) periodontal ligament (PDL) space, (6) Trabecular pattern, (7) cortex of the alveolar crest, (8) cortex of the mandibular canal, (9) furcation, (10) cortex of the mandible, and (11) overall image quality for periapical lesion diagnosis. Participants rated their responses on a six-point scale: (1) strongly disagree, (2) disagree, (3) slightly disagree, (4) slightly agree, (5) agree, and (6) strongly agree. 
To my point of view, that are not primary endodontic pattern and I miss a concrete endodontic diagnosis that should be present.

Response:

In endodontic treatment, the most critical aspect is determining the number of canals. Additional features such as the second mesiobuccal canal in maxillary molars, distolingual canal in mandibular molars, and extra roots like the radix entomolaris cannot be entirely discerned with only 2D periapical views. Therefore, the evaluation parameters include: (1) the number of roots and (2) the number of root canals. Assessing endo-origin lesions is also crucial, where changes occur in the normal anatomical structures, (3) enamel-dentin junction, (4) lamina dura, (5) periodontal ligament (PDL) space, (6) trabecular pattern, (7) cortex of the alveolar crest, (8) cortex of the mandibular canal, (9) furcation, and (10) cortex of the mandible if lesions are present. Therefore, selecting these parameters is essential, and we included this information in the discussion section as it seems to be missing from the paper.

In addition, we modified the title as the reviewer suggested. The criteria could be the parameters for diagnosis of tooth. Thank you for the comment.

[Page 9 Line 259-267]

The various parameters for the evaluation of subjective image quality were selected for the following reasons. During the diagnosis of tooth, important information is obtained by assessing the number of roots and root canals. CBCT facilitates the easy detection of additional anatomical features such as the second mesiobuccal canal in maxillary molars, additional roots (e.g., radix entomolaris), or the C-shaped canal system in mandibular molars. Assessing endo-origin lesions is also crucial, where changes occur in various anatomical structures, such as enamel-dentin junction, lamina dura, PDL space, trabecular pattern, cortex of the alveolar crest, cortex of the mandibular canal, furcation, and cortex of the mandible, if inflammation is present.

Comment #2:

The use of phantom heads may be adequate, but what is about simulating moving artifacts and all the other artifacts that are patient associated?

Response:

We agree with the reviewer’s opinion. In general, we use phantom head to analyze the quality of radiological images. However, phantom head for radiological evaluation does not have metal restorations and other causes of artifacts.

If we were to conduct the study on real patients, taking into consideration moving and other artifacts, maintaining uniformity in conditions for the analysis of radiological images would be impossible and there would be ethical issues about excessive radiation exposure to the same patient. I hope you understand these circumstances. Thank you for the comment.

Comment #3:

I question clinical relevance, as the mandible did not offer any typical endodontic diagnosis. 

Why was the evaluation limited to only one tooth? Why did you use different sizes of the FOV? You should better address the clinical necessity of larger FOVs – if present. I guess, the more information with the same or even lower doses, the better. 

To my point of view, the study simple highlighted the benefit of some AI-processing procedures but did not aim at an endodontic topic. For me, it is a native mandible and the AI processing led to dose reduction – please, do not use endodontic diagnoses masking the primary aim.

Response:

Most dental CBCT equipment have additional mode (D-mode) for diagnosis of single tooth and endodontic treatment. Hence, when we require diagnosis of a single tooth for endodontic treatment, we do not need to take large-FOV images. If we take large-FOV images for only a single tooth, it causes unnecessary radiation exposure to the patients. Moreover, each additional mode of CBCT machines has a different FOV, which cannot be adjusted by users.
In this study, we used the additional mode ‘D-mode’ for identifying the effectiveness of AI processed low-dose CBCT. This mode is usually used for the diagnosis of a single tooth. We modified the title and other parts as the reviewer suggested. Thank you for your comments.

[Page 1 Line 2-3]

Preclinical evaluation of AI-processed low-dose CBCT for the diagnosis of a single tooth

[Page 2 Line 48-56]

Indiscriminate use of cone-beam computed tomography (CBCT) to obtain anatomical information for tooth diagnosis should be avoided. Initial clinical examination and conventional radiography, such as periapical images, should be conducted. CBCT should be performed only when the clinical benefits outweigh the risks associated with radiation exposure [7]. In relation to single tooth diagnosis, suitable field-of-view (FOV) modes in CBCT may be considered in several situations. First, CBCT can detect pathological changes in periapical areas earlier than conventional 2D radiography. Second, in cases where anatomical complexity presents challenges, such as dens invaginatus and severely curved root canals (e.g., radix entomolaris), pretreatment imaging is beneficial.

Comment #4:

The title should be changed and you should focus on the true aim – AI processing and dose reduction. The study does not answer the question of “Clinical effectiveness”, as mentioned in the title. To prove clinical effectiveness, the AI processing should be performed in real clinical cases – this would be more relevant.

Response:

We agree with the reviewer’s opinion. We have changed the title as the reviewer suggested.

[Page 1 Line 2-3]

Preclinical evaluation of AI-processed low-dose CBCT for the diagnosis of a single tooth

Comment #5:

Recent publications addressed the topic even better: 
Lee, J. H., Lee, N. K., Zou, B., Park, J. H., & Choi, T. H. (2024). Reliability of Artificial Intelligence-based Cone Beam Computed Tomography Integration with Digital Dental Images. Journal of visualized experiments : JoVE, (204), 10.3791/66014. https://doi.org/10.3791/66014
Kim, K., Lim, C. Y., Shin, J., Chung, M. J., & Jung, Y. G. (2023). Enhanced artificial intelligence-based diagnosis using CBCT with internal denoising: Clinical validation for discrimination of fungal ball, sinusitis, and normal cases in the maxillary sinus. Computer methods and programs in biomedicine, 240, 107708. https://doi.org/10.1016/j.cmpb.2023.107708
Kim, K. S., Kim, B. K., Chung, M. J., Cho, H. B., Cho, B. H., & Jung, Y. G. (2022). Detection of maxillary sinus fungal ball via 3-D CNN-based artificial intelligence: Fully automated system and clinical validation. PloS one, 17(2), e0263125. https://doi.org/10.1371/journal.pone.0263125

Response:

We have added more references as the reviewer suggested. Thank you for recommendation.

[Page 2 Line 78-84]

However, more recently, the advent of artificial intelligence (AI)-based post-processing denoising solutions has shown promising results in further enhancing image quality.

The rapid growth of deep learning, particularly convolutional neural networks (CNNs), has revolutionized CBCT procedures by reducing radiation risks. CNNs automatically update features using input data and enhance image analysis accuracy for early detection of abnormalities.

Reviewer 2 Report

Comments and Suggestions for Authors

Manuscript Title “Clinical effectiveness of AI-processed low-dose CBCT for endodontic treatment”

General comment:

the quality of this manuscript is so so, since it cannot provide a correct optimal method in testing the experimental protocol plus the measured data do not have a appropriate interpretation to conclude the practical derivation. Thus, it needs a large revision to enhance the statements

Specific comment:

1.      Abstract: no solid result of improvement was listed

2.      Introduction: The length needs to be shortened. Either background or rationale study is not the major topic in the article, therefore, just long enough to lead the reader into the correlated topic and propose the theoretical basis of the method in quite enough.

3.      Materials and methods; no specific optimal method was introduced in this study to organize the protocol groups. It seems just randomly pick 4 protocols then compare to the standard. Suggest to adopt the Taguchi methodology or anyone else with five factors; FOV, voxel, exposure time, kVp and mA; and each has two level; thus 25=32 if using full factorial experimental method, or, skip one factor then take Taguchi into 8 combinations for surveying.

4.      Sec 2.3, the description of AI processing is too poor to imply the technique

5.      Sec 2.4. it has the same drawback as mentioned in sec 2.3

6.      Tab 3, no solid conclusion and further interpretation, just a list of derived data.

7.      Discussion, better add the subsection head to intensify the topic of this specific section, plus, there are no specific data to support the statements in the section. The evaluated dose should be the topic of this manuscript; however, I do not see any specific interpretation of the dose in the discussion.

8.      Conclusion, too short to express the result and essential points of discussion, try to extend to ~150 words.

Author Response

Point-to-Point Response to the Comments of Reviewer (bioengineering-2974874)

We are grateful to the reviewers for their critical comments and valuable suggestions that have helped us improve our manuscript considerably. As indicated in the following responses, we have incorporated all these suggestions into the revised version of our paper. The edited sentences are written in red in the revised manuscript. The page and line numbers are indicated in accordance with the manuscript in the Microsoft Word file.

[Reviewer 2]

Comment #6:

Abstract: no solid result of improvement was listed

Response:

According to your advice, we have revised the abstract.

[Page 1 Line 20-32]

This study assessed AI-processed low-dose cone-beam computed tomography (CBCT) for the diagnosis of a single tooth. Human-equivalent phantoms were used to evaluate CBCT image quality with a focus on the right mandibular first molar. A total of two CBCT were used for evaluation. The first CBCT was the experimental group, which was acquired in a total of four protocols and further enhanced with AI processing to improve image quality. The other machine was used as the control group, and the images were taken in one protocol without AI processing. The dose-area product (DAP) was measured for each protocol. Subjective clinical image quality was assessed by five dentists using 11 parameters and a six-point rating scale. The AI-processed protocols exhibited lower DAP/FOV (field of view) values than non-processed protocols, while demonstrating subjective clinical evaluation results comparable to those of non-processed protocols. These findings highlight the importance of minimizing radiation exposure while maintaining diagnostic quality as the usage of CBCT increases in single tooth diagnosis.

Comment #7:

Introduction: The length needs to be shortened. Either background or rationale study is not the major topic in the article, therefore, just long enough to lead the reader into the correlated topic and propose the theoretical basis of the method in quite enough.

Response:

According to your advice, we have shortened the Introduction section. The detailed explanation about situations where CBCT imaging is necessary during endodontic treatment appeared unnecessary, so it was shortened.

Comment #8:

Materials and methods; no specific optimal method was introduced in this study to organize the protocol groups. It seems just randomly pick 4 protocols then compare to the standard. Suggest to adopt the Taguchi methodology or anyone else with five factors; FOV, voxel, exposure time, kVp and mA; and each has two level; thus 25=32 if using full factorial experimental method, or, skip one factor then take Taguchi into 8 combinations for surveying.

Response:

The experimental group equipment we used in this study offers only two modes (standard mode and fast mode). The fast mode is to acquire images in a shorter (half) time with half the radiation exposure time of standard mode. Therefore, the radiation exposure is half as high and image quality is lower. In addition, we could add filters to the equipment to reduce the radiation dose. Filters also reduce the radiation dose but also cause a reduction in image quality. Other values are generally unchanged for clinical use of CBCT. Exposure time, kVp, and mA can be changed, but most clinicians use the preset modes (standard or fast).

Using filters, fast mode, and additional AI processing, there were a total of eight protocols in this study and we believe this experimental group the most appropriate one to conduct this study. We will definitely consider your suggestions when planning further studies. Thank you for the comment.

Comment #9:

Sec 2.3, the description of AI processing is too poor to imply the technique

Response:

We modified the description of AI processing as the reviewer suggested. Thank you for comment.

[Page 4 Line 143-157]

The EDCNN was used for denoising of low-dose CT images. Figure 3 depicts the architecture of the network. First, the edge enhancement module consisting of the trainable Sobel convolutions (eight groups of the four types) extracted edge information and enriched the input information. The Sobel convolution was designed with a learnable Sobel parameter in the form of the traditional Sobel operator. Accordingly, 32 different edge features were extracted using this module. At the last stage of the module, low-dose CT images were stacked with the extracted edge features in the channel dimension. Next, the dense connection module was composed of 1×1 and 3×3 convolutions and Leaky ReLU layers. To maximize the use of the extracted edge features and original input, edge features extracted from the edge enhancement module were connected by skip connection to each B module. The output channel of the module remained consistent at 32. Through point-wise convolution with a 1×1 kernel, the channel size was converted to 1, aligning it with the input channel. The convolutional network with a padded 3×3 kernel was used to adjust the output spatial size to the input size. Finally, denoised images were obtained by adding the original low-dose CT images via skip connection.

Comment #10:

Sec 2.4. it has the same drawback as mentioned in sec 2.3

Response:

We have added more description to Sec 2.4. Thank you for the comments

[Page 5 Line 175-179]

Images were presented to the evaluators using the Rainbow 3D viewer 1.0.0 (Dentium, Suwon, Korea), which allowsaxial, coronal, and sagittal views of CBCT images for evaluation (Fig. 4). This is the same program used to capture actual CBCT images in clinical settings and for diagnostic evaluation. To ensure fairness, an observation time of 3 minutes was allotted for each image.

Comment #11:

Tab 3, no solid conclusion and further interpretation, just a list of derived data.

Response:

We have added more explanation about the data in table 3.

[Page 7 Line 202-210]

The lowest score was attributed to protocol 4 images without AI processing (score order: 1 AI, 2 AI, 5, 1, 3 AI, 2, 3, 4 AI, and 4). Incorporating AI processing into the experimental group images resulted in an overall enhancement of the subjective clinical evaluation scores. AI-processed protocol 2 images showed higher scores than non-processed protocol 1 images. AI-processed protocol 3 images showed higher scores than non-processed protocol 2 images. AI-processed protocol 4 images showed a higher score than non-processed protocol 3 images. Even though AI-processed protocol 2 images had lower DAP per FOV than protocol 5 images, AI-processed protocol 2 images showed higher scores than protocol 5 images (Table 3).

Comment #12:

Discussion, better add the subsection head to intensify the topic of this specific section, plus, there are no specific data to support the statements in the section. The evaluated dose should be the topic of this manuscript; however, I do not see any specific interpretation of the dose in the discussion.

Response:

We feet that creating a subsection in the Discussion would make the content disjointed; hence, we apologize that we don't agree with creating a subsection in the Discussion. Moreover, the most important aspect of our study is the improvement of image quality of low-dose CBCT by AI processing, so we added more content about the interpretation of the results to the Discussion. Thank you for comment.

[Page 9 Line 282-287]

In addition, the subjective image evaluation in the experimental group showed that AI-processed protocol 2 images showed a higher score than non-processed protocol 1 images. AI-processed protocol 3 images showed a higher score than non-processed protocol 2 images. AI-processed protocol 4 images showed a higher score than non-processed protocol 3 images. This confirms that AI processing can improve radiological image quality by more than one level.

[Page 10 Line 298-302]

The findings of this study are significant, as AI-processed low-dose CBCT in the experimental group demonstrated subjective image quality evaluation results that were clinically comparable to those of conventional CBCT in the diagnosis of a single tooth. However, further developing AI-processing technology to improve image quality is important.

Comment #13:

Conclusion, too short to express the result and essential points of discussion, try to extend to ~150 words.

Response:

According to your advice, we have revised the conclusion.

[Page 10 Line 320-330]

This research concluded that incorporating AI into low-dose CBCT image processing is valuable in clinical practice, significantly enhancing subjective image quality. The study's findings validate the use of low-dose CBCT for the diagnosis of a single tooth, relying on subjective image assessment with key parameters. By utilizing this method, precise diagnosis can be achieved with reduced radiation exposure, easing the burden of extensive examinations for patients, especially for those whom radiation exposure is a concern, such as adolescents and individuals with systemic conditions. These results can serve as a pivotal basis for future studies and contribute to the advancement of safe and efficient image enhancement through AI technologies, both in clinical practice and from an ethical perspective.

Round 2

Reviewer 1 Report

Comments and Suggestions for Authors

Dear Authors,

Thank you for resubmitting the revised version of your manuscript.

The quality improved, but there are still some major aspects that need explanation.

The study included the analysis of one native tooth without any pathological finding and you evaluated subjective findings of five investigators. The process was performed once and investigators had a 6-point Likert scale for rating.

I cannot see any analysis concerning the inter- and intra-rater reliability because this would require a second assessment of the subjective image quality evaluation.

I strongly recommend these data. The statistical analysis is very poor, as you only compared mean values of all investigators.

Additionally, I miss a proper statistical analysis. I cannot find any adequate statistical analysis and the corresponding p-values that may show significances.  

The analysis included non-parametric values – thus, please provide a proper statistical analysis.

Your study only represents a descriptive presentation of some AI-processes in CBCT imaging.

I strongly recommend to rethink about your study design and offer a better design and more meaningful results. In its current form, I cannot accept the manuscript.

What is the finding based on subjective image quality evaluation that was performed once without checking inter- and intra-rater reliability?

Best wishes

Comments on the Quality of English Language

minor English editing is required

Author Response

Thank you so much for your feedback. The authors put a lot of thought into improving the reliability of this study. Per your suggestion, we agree that it was necessary to analyze the inter-reader reliability. Accordingly, we conducted additional statistical analyses using Fleiss' kappa statistics. We hope to improve the impact of our findings with this additional analysis.

We have added additional statistical methods and results to the text and added a discussion about them. We sincerely appreciate the reviewer's suggestions.

[Page 6 Line 186-194]

2.5 Statistical Analysis

SPSS (version 20, IBM Corp) was used for statistical analyses. Inter-reader reliability of parameter average was assessed using Fleiss' kappa statistics. The Fleiss’ kappa coefficient was applied to assess agreement and statistical significance (<0.00, poor agreement; 0.00–0.20, slight agreement; 0.21–0.40, fair agreement; 0.41–0.60, moderate agreement; 0.61–0.80, substantial agreement; and 0.81–1.00, almost perfect agreement). The Fleiss’ kappa coefficient > 0 indicates progressively better-than-chance agreement among raters, with a maximum value of +1 signifying perfect agreement. P < 0.05 was considered to indicate statistical significance[22, 23].

[Page 7 Line 223-225]

The Fleiss' kappa coefficient of parameter average was 0.619 and was statistically significant (p = 0.031). The Fleiss' kappa coefficient showed substantial agreement.

[Page 9 Line 289-296]

In addition, this study used subjective image quality evaluation to evaluate each protocol. Since there is no way to objectively and accurately judge image quality, we believe that it is meaningful for clinicians who evaluate radiographic images in clinical practice to evaluate each protocol by themselves. This study also used Fleiss's kappa statistics to check inter-reader reliability, which revealed that the clinicians who participated in the evaluation showed statistically significant, substantial agreement. These results show that the subjective image quality evaluation of this study did not differ significantly among the raters.

Reviewer 2 Report

Comments and Suggestions for Authors

Manuscript Title “Clinical effectiveness of AI-processed low-dose CBCT for endodontic treatment”

General comment:

The revised version is much better than the last one. It is OK to accept. Sec 2.3 provides more technical description in the revised version but still no subsection is added in the discussion to emphasize the specific topic in every section. the revised conclusion is suitable to imply a strong ending.

Author Response

Thank you for your positive response. Per your recommendation, we considered creating a subsection in the discussion. After careful evaluation, we determined that the content was better presented in its current format. Thank you for your understanding in this matter, which we appreciate.  

Round 3

Reviewer 1 Report

Comments and Suggestions for Authors

Dear Authors,

Thank you for resubmitting the next revised version of your manuscript.

The quality improved, but there are still some major aspects that need explanation.

The study included the analysis of one native tooth without any pathological finding and you evaluated subjective findings of five investigators. The process was performed once and investigators had a 6-point Likert scale for rating.

You added some statistical analysis and offered Fleiss-Kappa for the inter-rater reliability. This is a nice way to overcome the missing analysis and the poor study design.

Nonetheless, I must state, that the statistical analysis is still poor. Why did you not perform the analysis twice? Do the different AI-processed CBCT diagnosis offer an intra-rater reliability? You did not answer this question.  The second evaluation is still missing and your analysis has a descriptive character. You should highlight the limitations of the study more clearly.

The following sentence remain

I strongly recommend to change the title in “Preclinical and preliminary evaluation of perceived image quality of AI-processed low-dose CBCT analysis of a single tooth”

The study is very helpful as the CBCT analysis may require less radiation dose in future.

I do not want to be the killjoy.

The study is based on a good idea. Nonetheless, your setup/design is poor and the statistical analysis, too. I still can't get anything out of the study design without real clinical reference and real pathological findings. It is on the editor!

I still think the study does not justify a publication in such a high ranked journal, as mentioned before.

Best wishes

Comments on the Quality of English Language

minor English editing is required

Author Response

Comment #1:

The quality improved, but there are still some major aspects that need explanation. 

The study included the analysis of one native tooth without any pathological finding and you evaluated subjective findings of five investigators. The process was performed once and investigators had a 6-point Likert scale for rating. 

You added some statistical analysis and offered Fleiss-Kappa for the inter-rater reliability. This is a nice way to overcome the missing analysis and the poor study design. 

Nonetheless, I must state, that the statistical analysis is still poor. Why did you not perform the analysis twice? Do the different AI-processed CBCT diagnosis offer an intra-rater reliability? You did not answer this question.  The second evaluation is still missing and your analysis has a descriptive character. You should highlight the limitations of the study more clearly. 

Response:

Thank you so much for your feedback. We performed a second subjective clinical image evaluation to assess intra-rater reliability. Based on the results, we have modified all the data, including the average of the scores and the inter-rater reliability. In addition, we have supplemented the limitations of this study with more details in the Discussion section. We hope that our additional analyses and efforts will meet the reviewer’s standards.

[Page 1 Lines 26-29]

Subjective clinical image quality was assessed twice by five dentists, with a 2-month interval in between, using 11 parameters and a six-point rating scale. Agreement and statistical significance were assessed with Fleiss’ kappa coefficient and intra-class correlation coefficient.

[Page 1 Lines 32-33]

The intra-class correlation coefficient showed statistical significance and almost perfect agreement.

[Pages 5-6 Line 184-186]

To investigate intra-rater reliability, subjective clinical image quality evaluations were performed twice by the same evaluators, with a 2-month interval in between, to allow for a washout period.

[Page 6 Lines 198-203]

Intra-rater reliability was assessed using intra-class correlation coefficient (ICC). A six-tiered classification system was utilized to evaluate the level of agreement: poor agreement (ICC, < 0.00), slight agreement (ICC, 0.00–0.20), fair agreement (ICC, 0.21–0.40), moderate agreement (ICC, 0.41–0.60), substantial agreement (ICC, 0.61–0.80), and almost perfect agreement (ICC, 0.81–1.00). All p values < 0.05 were considered statistically significant [24].

[Page 7 Lines 232-237]

The Fleiss' kappa coefficient of parameter averages was 0.621, which was statistically significant (p = 0.030). The Fleiss' kappa coefficient showed substantial agreement. Additionally, the ICC value of 0.995 indicated an almost perfect agreement in intra-rater reliability. The ICC value ranged from 0.976 to 0.998 within the 95% confidence interval. The p-value was < 0.001, demonstrating statistically significant results.

[Page 9 Lines 310-313]

This study also used ICC statistics to assess intra-rater reliability, which showed statistical significance and almost perfect agreement. These results show that the subjective image quality evaluation in this study did not differ significantly in the inter- and intra-rater evaluations.

[Page 10 Lines 347-350]

Moreover, inter- and intra-reliability assessments were performed in this study to ensure the reliability of subjective ratings. While this may still be insufficient, we acknowledge it as an unavoidable limitation of the method used to assess actual clinical effectiveness.

[Page 10 Lines 353-357]

However, ensuring the standardization of examination can be difficult if the skull contains pathologic problems or abnormal anatomical structures. A phantom skull with normal findings would be more appropriate for evaluating the quality of radiographic images. Hence, a phantom skull was used in this study.

Comment #2:

I strongly recommend to change the title in “Preclinical and preliminary evaluation of perceived image quality of AI-processed low-dose CBCT analysis of a single tooth” 

The study is very helpful as the CBCT analysis may require less radiation dose in future. 

I do not want to be the killjoy. 

The study is based on a good idea. Nonetheless, your setup/design is poor and the statistical analysis, too. I still can't get anything out of the study design without real clinical reference and real pathological findings. It is on the editor!

I still think the study does not justify a publication in such a high ranked journal, as mentioned before. 

Response:

Thank you so much for your feedback. We have modified the Title as the reviewer suggested. In addition, the quality of our paper has improved thanks to the reviewers’ comments. The purpose of evaluating the degree of image quality improvement with AI features is to assess how useful the AI processed images will be in clinical practice. Therefore, we believe that this study is adequate for this purpose. While we understand the reviewer's comments, we hope that the reviewer understands that there are inevitable limitations to how we evaluate the degree of image improvement.

[Page 1 Lines 2-4]

Preclinical and preliminary evaluation of perceived image quality of AI-processed low-dose CBCT analysis of a single tooth
